# Effects of Gut Microbiome Modulation on Reducing Adverse Health Outcomes among Elderly and Diabetes Patients during the COVID-19 Pandemic: A Randomised, Double-Blind, Placebo-Controlled Trial (IMPACT Study)

**DOI:** 10.3390/nu15081982

**Published:** 2023-04-20

**Authors:** Martin C. S. Wong, Lin Zhang, Jessica Y. L. Ching, Joyce W. Y. Mak, Junjie Huang, Shilan Wang, Chris K. P. Mok, Angie Wong, Oi-Lee Chiu, Yee-Ting Fung, Pui-Kuan Cheong, Hein-Min Tun, Siew C. Ng, Francis K. L. Chan

**Affiliations:** 1The Jockey Club School of Public Health and Primary Care, Faculty of Medicine, The Chinese University of Hong Kong, Hong Kong SAR, China; 2Centre for Health Education and Health Promotion, Faculty of Medicine, The Chinese University of Hong Kong, Hong Kong SAR, China; 3Microbiota I-Center (MagIC), Hong Kong SAR, China; 4Centre for Gut Microbiota Research, Faculty of Medicine, The Chinese University of Hong Kong, Hong Kong SAR, China

**Keywords:** gut microbiota, microbiome immunity formula, adverse health outcomes, gut dysbiosis, quality of life, vulnerable populations

## Abstract

Gut microbiota is believed to be a major determinant of health outcomes. We hypothesised that a novel oral microbiome formula (SIM01) can reduce the risk of adverse health outcomes in at-risk subjects during the coronavirus disease 2019 (COVID-19) pandemic. In this single-centre, double-blind, randomised, placebo-controlled trial, we recruited subjects aged ≥65 years or with type two diabetes mellitus. Eligible subjects were randomised in a 1:1 ratio to receive three months of SIM01 or placebo (vitamin C) within one week of the first COVID-19 vaccine dose. Both the researchers and participants were blinded to the groups allocated. The rate of adverse health outcomes was significantly lower in the SIM01 group than the placebo at one month (6 [2.9%] vs. 25 [12.6], *p* < 0.001) and three months (0 vs. 5 [3.1%], *p* = 0.025). At three months, more subjects who received SIM01 than the placebo reported better sleep quality (53 [41.4%] vs. 22 [19.3%], *p* < 0.001), improved skin condition (18 [14.1%] vs. 8 [7.0%], *p* = 0.043), and better mood (27 [21.2%] vs. 13 [11.4%], *p* = 0.043). Subjects who received SIM01 showed a significant increase in beneficial Bifidobacteria and butyrate-producing bacteria in faecal samples and strengthened the microbial ecology network. SIM01 reduced adverse health outcomes and restored gut dysbiosis in elderly and diabetes patients during the COVID-19 pandemic.

## 1. Introduction

Worldwide, the coronavirus disease 2019 (COVID-19) pandemic has posed a substantial challenge in terms of its induced morbidity and mortality to the general population [1]. Patients with diabetes and elderly individuals are particularly vulnerable during the pandemic. They are not only more susceptible to various infections but also develop more severe illnesses, as compared to subjects without diabetes and the young population [2,3]. Strategies to protect these two groups of vulnerable subjects, which are particularly prone to develop adverse health outcomes during the pandemic, represent a priority in healthcare policy. The reasons for a higher rate of adverse health outcomes among patients with underlying type two diabetes mellitus (DM) are multifactorial. First, diabetic patients are more likely to be older and have other comorbidities. Furthermore, diabetic patients may have alterations in their innate immune system that promote proinflammatory cytokines [4,5,6,7]. In addition, elderly people are also predisposed to infections and their complications [8]. Patients older than 65 years of age had a 23-fold greater risk of mortality than those younger than 65 [9]. They are characterised by high proinflammatory immune reactions and low adaptive immune responses [10,11]. With an ageing population and a high prevalence of type two diabetes mellitus, these individuals represent a large vulnerable group amidst the COVID-19 pandemic. Thus, there is an urgent need to implement strategies to reduce adverse health outcomes among these individuals.

Accumulating evidence suggests that gut microbiota plays an important role in human health, including digestive functions and the risk of sepsis [12]. Gut dysbiosis could alter the functions of innate and adaptive immunity, leading to increased susceptibility to various diseases and infections [13,14]. In the past decade, major prebiotic compounds, such as xylooligosaccharides, were found to be effective in enhancing the growth and activity of gut symbionts [15], such as Lactobacillus, Bifidobacterium, and Anaerostipes spp. In addition, studies have shown that probiotics regulate the functions of systemic and mucosal immune cells and intestinal epithelial cells [16]. In an open-label pilot study, we have shown that a novel gut microbiota-derived synbiotic formula, SIM01, hastened antibody formation against SARS-CoV-2, reduced blood proinflammatory immune markers, and restored gut dysbiosis in patients hospitalised for acute COVID-19 [17]. We have also identified that certain beneficial bacteria such as *Bifidobacterium adolescentis*, Prevotella copri, and two Magamonas species have anti-inflammatory effects and are associated with fewer adverse health events after vaccination [18]. In our recent local population-based study, elderly people and patients with type two diabetes mellitus have gut dysbiosis with a deficiency of beneficial bacteria, including Bifidobacteria [19]. The latter may, in part, account for the increased risk of adverse health outcomes among these patients, especially during the COVID-19 pandemic.

To date, large-scale, randomised trials are lacking to evaluate the clinical benefits of probiotics among at-risk individuals. The Immune Microbiome Product Against COVID infecTion (IMPACT) study aimed to assess the efficacy of a novel microbiome immunity formula (SIM01) in reducing adverse health outcomes in the elderly and patients with type two diabetes mellitus during the COVID-19 pandemic. We also assessed whether the use of SIM01 could improve quality of life and gut dysbiosis.

## 2. Materials and Methods

### 2.1. Study Design and Participants

This was a double-blind, randomised, parallel-arm, placebo-controlled trial conducted in a university centre in Hong Kong. Subjects were recruited from a territory-wide, community-based recruitment initiative that reached the general population of Hong Kong from April 2021 to March 2022. Eligible subjects consisted of individuals aged ≥65 years and patients with diabetes mellitus who were naive to COVID-19 vaccination. Subjects were excluded if they had: a history of confirmed COVID-19 infection; active sepsis or active malignancy; underlying immunosuppressed state such as a prior organ or hematopoietic stem cell transplant; neutropenia with absolute neutrophil count (ANC) < 500 cells/μL at the time of study inclusion; HIV infection with CD4 < 200 cells/μL at the time of study inclusion; concomitant use of immunosuppressants or corticosteroids at a prednisolone-equivalent dose of ≥10 mg for more than three months; history or active infective endocarditis; been on peritoneal dialysis or haemodialysis; positive pregnancy test; or used prebiotics, probiotics, or antibiotics within one month prior to the study. Patients with diabetes mellitus should have had stable disease control and no changes in medications for ≥3 months. The study was approved by the Clinical Research Ethics Committee of the Chinese University of Hong Kong on 19 April 2021 (Ref. No. 2021.186). It was registered on ClinicalTrials.gov (identifier NCT04884776). This study was conducted according to the principles of the Declaration of Helsinki and ICH-GCP. All subjects provided informed consent.

### 2.2. Randomisation and Masking

After the screening phase, study subjects were randomised by computer-generated numbers to receive a microbiome immunity formula (SIM01) or placebo in a 1:1 ratio for three months (Figure 1). Eligible subjects were randomised at the time of COVID-19 vaccination to reduce the confounding effects of vaccination on health outcomes. All subjects received the COVID-19 vaccine, including either the inactivated vaccine Sinovac-CoronaVac or the mRNA vaccine BNT162b2 (Pfizer-BioNTech), within one week of intervention. Study visits were arranged at 1 month and 3 months after completion of the second dose of the COVID-19 vaccine. SIM01 is an oral microencapsulated formulation of three lyophilised Bifidobacteria at a dose of 20 billion colony-forming units (CFU) per day and 3 prebiotics, including galactooligosaccharides, xylooligosaccharide, and resistant dextrin, as derived from our previous study [20,21]. The three Bifidobacterium strains (*B. adolescentis*; *B. bididum*, and *B. longum*) are commercially available from Chambio Co., Ltd. and WECARE-PROBIOTICS. SIM01 has been approved by health authorities and is a patent-protected formula. The placebo consisted of 2 mg of vitamin C per day and was composed of an inert substance, which was made of starch filler, flavour and colouring. Its appearance and taste were identical to the study product. To ensure masking, packaged placebo products were dispensed to the patients by a separate investigator who was unaware of the treatment allocation. All study subjects and investigators were blinded to the randomisation codes, which were kept confidential until the completion of all follow-up visits. A computer-generated randomisation schedule was used to assign subjects to the treatment sequences. To ensure allocation concealment, an independent staff dispensed consecutively numbered, identically designed treatment packs that contained sealed bottles of the study products. The SIM01 formula and placebo were produced in GMP-accredited facilities and were stored at, and dispensed by, a central research pharmacy. Incentives were provided to the recruited subjects at each study visit. A governance system was in place to monitor the disbursement of incentives to ensure accountability and traceability.

### 2.3. Study Visits

At the baseline visit, sociodemographic information, clinical details, and history of COVID-19 vaccination were obtained from study subjects or extracted from their medical notes. Faecal specimens were collected for microbiota analysis. They were provided with study packages during the visit. Study participants were followed up in a research clinic with faecal specimen collection for metagenomic sequencing at 1 month and 3 months after the second dose of the vaccine. Stool specimens were tested for microbial analysis by metagenomic sequencing. We documented any hospital admission in each follow-up visit. Compliance with taking the SIM01/placebo was assessed by returned sachet pack counts at 1 and 3 months. No probiotic or prebiotic preparations were permitted during the study period. The use of antibiotics should be avoided unless there was a clinical need. Study participants with an intake of prohibited medication or antibiotics during the study period remained in the study and the outcome was assessed. The reported intake of prohibited medication or antibiotics was recorded. We documented the number and reasons for early withdrawals.

### 2.4. Outcome Measures

The primary outcome was a composite of adverse health outcomes, defined as new symptoms or diseases leading to self-reported negative impacts on their activities of daily living including hospitalisation, absence from work, and medical consultations.

The collection of adverse health outcomes commenced from the time the subjects were enrolled in the study. Secondary outcomes were changes in quality of life and gut microbiota composition. We examined whether study subjects had improvement in quality of life regarding their sleep quality, skin appearance, and mood three months after the intervention. Three questions were constructed by a public health professional and pilot tested by an epidemiologist for face validity. These questions included: in the past three months, (i) was your sleep quality better, worse, or similar; (ii) did your skin condition appear younger, older, or similar; and (iii) was your mood better, worse, or similar? We also measured the restoration of gut dysbiosis, defined as improvement in gut microbiome composition and diversity, and the proliferation of a beneficial bacteria genus (i.e., Bifidobacterial and other short-chain fatty acids producers).

### 2.5. Stool Metagenomic Sequencing and Profiling

Faecal samples were collected at baseline and at three months after the second dose of COVID-19 vaccination (Figure 2A). We performed metagenomic profiling on faecal samples, as previously described [18]. Briefly, faecal DNA was extracted using a Maxwell^®^ RSC PureFood GMO and Authentication Kit (Promega, Madison, WI, USA) following the manufacturer’s instructions. Sequencing libraries were prepared from extracted DNA using the Illumina^®^ DNA Prep, (M) Tagmentation Kit (Illumina, San Diego, CA, USA), and sequenced with a paired-end 150 bp sequencing strategy by Illumina NovaSeq 6000 System, generating 93.5 ± 15.2 million (mean ± SD) raw reads per sample. Raw sequence reads were filtered and quality trimmed using a Trimmomatic v0.9 and decontaminated against the human genome (Reference: hg38) by Kneaddata (V. 0.10.0, https://bitbucket.org/biobakery/kneaddata/wiki/Home, accessed on 1 February 2023). MetaPhlAn3 (v 3.0.13) was used to generate a species-level abundance table. Alpha microbial diversity was studied using the Chao1 richness index and the Shannon diversity index based on the species-level profile for each sample. Beta diversity was assessed with Jensen-Shannon Divergence (JSD) from phyloseq and vegan packages and visualized by nonmetric multidimensional scaling (NMDS).

### 2.6. Statistical Analysis

The sample size estimation was to detect a sizeable effect size of reduction in adverse health outcomes by the intervention. Based on previous literature on the incidence of adverse outcomes in diabetes patients and elderly populations [22,23,24], we assumed that the incidence of a composite of adverse health outcomes was 10% and 3%, respectively, in the control subjects and those in the SIM01 group. In two-sided tests with α = 0.05, a total of 444 subjects were needed to achieve a power of 80%.

Continuous variables were expressed in the mean (±standard deviation) while categorical variables were presented as a percentage. Changes in continuous variables, including the quantities of bacteria, were compared by the Wilcoxon rank-sum test, whereas changes in categorical variables were compared using the Chi-square test or Fisher’s exact test. A two-sided *p* value of <5% was considered to be statistically significant.

Fold change (month 3/baseline) of species abundance within each subject was calculated first and then Wilcoxon rank-sum tests (two sides) were performed to compare differences between different groups. Paired Wilcoxon rank-sum tests (two sides) were performed to compare the alpha diversity, and COVID (Long COVID)-positive correlation bacterial species of baseline and month 3 samples within each treatment group. Pairwise multilevel comparisons among the baseline and month 3 samples of the placebo and SIM01 treatment group were carried out on the JSD distance matrix using the pairwise Adonis test, which was assessed using permutational multivariate analysis of variance (PERMANOVA). The differential species between baseline and month 3 samples within each treatment group were evaluated with MaAsLin2 analyses, considering subject ID as a random effect. The significance threshold for these analyses was a false discovery rate (FDR)-corrected *p*-value (q-value) of 0.25. Differentially abundant species at month 3 between the adverse health outcomes group or the COVID-19 groups were identified using the linear discriminant analysis effect size (LEfSe v1.1.01) (linear discriminant analysis (LDA) score > 2, and *p* < 0.05). The correlation between species abundance was analysed using Spearman’s correlation tests. All microbiome-related statistical tests were performed with R Statistics version 4.1.3 with the following packages: phyloseq, vegan, tidyverse, dplyr, pairwise.adonis, ggplot2, and ggpubr.

### 2.7. Patient and Public Involvement

For this randomised controlled trial, no patients were directly involved in setting the research question, outcome measures, study design, or implementation. Study subjects attended follow-up visits and completed surveys. No patients were involved in the interpretation or writing up of the results.

## 3. Results

### 3.1. Participant Characteristics

Between April 2021 to March 2022, 497 subjects were screened (Figure 1). Forty-two subjects were excluded (not meeting eligibility criteria: 20; refusal or nonattendance: 22). Figure 1 shows the CONSORT flow diagram of subject recruitment and follow up at different phases. Finally, a total of 453 subjects were enrolled: 224 to SIM01 and 229 subjects to placebo (Table 1). The attrition rate of the intervention and control groups was 8% and 13.5% at one month and 14.7% and 16.2% at three months, respectively. The mean age was 67.5 years (SD: 8.1) with an equal gender ratio (female 50%). Among them, 49.7% were elderly and 50.3% were patients with type two diabetes. The majority were nonsmokers (84.7%) and nondrinkers (85.4%). The most common chronic conditions were hypertension (49.2%), diabetes (47.5%), hyperlipidaemia (47%), cardiovascular diseases (8.4%), and fatty liver (5.3%). 

### 3.2. Adverse Health Outcomes

The proportion of subjects with a combined composite of adverse health outcomes (2.9% vs. 12.6%, *p* = 0.0002) was significantly lower in the SIM01 group than in the placebo group at one month (Table 2). The proportion of subjects with these adverse outcomes (0% vs. 3.1%, *p* = 0.0252) also remained significantly lower in the SIM01 than in the placebo group at three months. Overall, 61.3%, 19.4%, and 19.4% of the adverse health outcomes at one month were gastrointestinal upset (bloating, constipation, diarrhoea, abdominal discomfort, and loss of appetite), immunological reactions (dermatitis and urticarial rash), and infections, respectively, leading to medical consultations or hospitalisations (Table 2). The corresponding proportion of adverse health outcomes at three months was 40%, 0%, and 60%, respectively. Seven subjects (3.1%) in the SIM01 and five subjects (2.2%) in the placebo group discontinued the study products or had poor compliance. A subgroup analysis excluding these subjects showed that the incidence of adverse health outcomes was also lower in the SIM01 group. The incidence of possible adverse events of COVID-19 vaccines was lower in the SIM01 group than in the control group at one month (2.9% vs. 9.6%) and three months (0% vs. 1.2%).

We conducted a subgroup analysis according to the patient groups (elderly people vs. diabetes patients). It was found that the incidence of adverse health outcomes at one month was significantly higher in the control group than the SIM01 group among diabetes patients (8.7% vs. 1.8%, *p* = 0.029) and the elderly people (16.8% vs. 4.3%, *p* = 0.008). A similar difference was detected in these two groups (3.8% vs. 0% and 2.5% vs. 0%, respectively) at three months.

Quality of life was significantly improved at the end of the intervention in the SIM01 group. The proportion of study participants who reported better sleep quality (41.4% vs. 19.3%, *p* < 0.001), improved skin appearance (14.1% vs. 7.0%, *p* = 0.043), and improved mood (21.1% vs. 11.4%) at three months was higher in the SIM01 than in the placebo group.

### 3.3. Faecal Microbiota Composition

At baseline, we observed no significant difference in the relative abundances of three probiotic Bifidobacterium species between the SIM01 and placebo groups (*p* > 0.05) (Appendix A). While the placebo group showed no significant differences in species richness and diversity, the SIM01 group showed a slight increase in the Shannon diversity at three months postvaccination (*p* = 0.061, Figure 2B,C). On the other hand, the overall microbial community of the SIM01 group at three months postvaccination was significantly different from that of the placebo group (*p* = 0.024, Figure 2D), along with increased relative abundances of the probiotic species contained in SIM01, as indicated by a higher aggregated fold change (*p* = 1.4 × 10^−10^; Figure 2E) or a higher fold change of individual probiotic species in the SIM01 group than the placebo group (*Bifidobacterium adolescentis*, *p* = 2.2 × 10^−6^; Bifidobacterium bididum, *p* = 0.00082; and Bifidobacterium longum, *p* = 0.011; Appendix A). Among these two enriched probiotic Bifidobacterium species, the abundance of *B. adolescentis* is associated with fewer adverse health outcomes at three months postvaccination and COVID-19 infection rate (Figure 3). Aside from these two Bifidobacterium species, the effect of SIM01 intervention could be noticeable by the specific enrichments of other beneficial species, including Bifidobacterium kashiwanohense, Bifidobacterium catenulatum, Bifidobacterium pseudocatenulatum, Blautia coccoides, and significant reductions of detrimental pathogens such as Bacteroides nordii and Bacteroides intestinalis (adjusted *p* < 0.05 vs baseline, Figure 4). To further demonstrate whether SIM01 administration is essential to restoring gut dysbiosis in subjects with vaccination, we defined pathogenic bacteria according to the faecal metagenomic analysis of the microbial profile, as previously described (Appendix A). Unlike the placebo group, SIM01 intervention significantly reduced the collective abundance of detrimental bacterial species that have been shown previously in association with COVID-19 or long COVID (*p* < 0.01, Figure 5).

We previously reported that gut microbiome composition may associate with adverse events caused by vaccination [18]. Consistently, our PAM clustering indicated that subjects in the SIM01 group with the enterotype enriched by *B. adolescentis* species at three months postvaccination had fewer adverse health outcomes and COVID-19 infection rates (Figure 3). Furthermore, the gut microbiota of subjects without adverse health outcomes had a higher abundance of Gemmiger formicilis and Oscilibacter sp_57_20 species, and a lower abundance of Bacteroides thetaiotaomicron at three months postvaccination (Figure 6A). From our correlation analysis, we found that *B. adolescentis* showed a significant positive correlation with Gemmiger formicilis and Oscilibacter sp_57_20 species while it was negatively correlated with B. thetaiotaomicron at three months postvaccination (Figure 6B).

## 4. Discussion

To the best of our knowledge, this is the first intervention trial showing that a novel microbiome formula, SIM01, could reduce adverse health outcomes, improve quality of life, and restore gut dysbiosis among elderly subjects and patients with type two diabetes during the COVID-19 pandemic. Using metagenomic sequencing, we demonstrated effective colonisation of probiotic species of SIM01 in the gut. Importantly, SIM01 not only replenished Bifidobacteria but also favoured the coexistence of other beneficial species. On the other hand, we also found that certain detrimental bacteria enriched in COVID-19 or long COVID patients were significantly reduced after receiving SIM01. Overall, our data suggested that SIM01 restored the commensal bacteria and suppressed detrimental pathogens in high-risk populations.

The strengths of this study include a rigorous randomised study design with controls that allows a robust evaluation of the modulation impact of the SIM01 formula on reducing the various study outcomes. Predefined endpoints were also used, including a composite of adverse health outcomes which exerted significant impacts on the activities of daily living of the study subjects. In addition, randomisation was conducted at the time of vaccination to minimise the confounding effects of the vaccines on health outcomes. Although ascertainment of these outcomes required subjective evaluation regarding their relationship with the intervention, the study involved only a single, blinded assessor to reduce bias. We also used metagenomic sequencing as objective measures to characterise both the intervention and control groups.

One of the reasons for the impaired immune response was due to the depletion of certain beneficial species within their gut microbiota, particularly *B. adolescentis*. A previous study also showed that supplementation with *B. adolescentis* can improve osteoporosis, neurodegeneration, and other age-related conditions in mice by regulating oxidative stress and inflammation [25]. Consistently, in this study, we also observed that the enrichment of *B. adolescentis* by SIM01 in vulnerable individuals was significantly associated with reduced risks of adverse health outcomes following vaccination.

Observational studies have shown that patients with COVID-19 infection reported altered gut microbiota composition, including reduced bacterial diversity, decreased abundance of short-chain fatty acid-producing bacteria from the Lachnospiraceae, Ruminococcaceae, and Eubacteriaceae families, as well as increased opportunistic pathogens from the Enterobacteriaceae families [26]. Nevertheless, oral administration of oral probiotics and prebiotics has been proven effective to induce antiviral activity and exert favourable effects on gut microbial diversity and composition which was associated with better clinical outcomes in COVID-19 patients [26]. Furthermore, a recent study in the US by Wischmeyer and colleagues examined the probiotic Lactobacillus rhamnosus GG (LGG) as postexposure prophylaxis for COVID-19 in 182 participants who had household exposure to someone with confirmed COVID-19 diagnosed within ≤7 days [27]. The findings suggested that LGG was associated with prolonged time to COVID-19 infection, reduced incidence of illness symptoms, and gut microbiome changes. However, the study was limited by a smaller-than-expected sample size, which limited its statistical power. Secondly, this US study has not included vaccinated individuals, which is an important limitation given the high transmissibility of newer viral strains and the potential for waning vaccine efficacy.

Our study had limitations. First, it has been shown that microbiome modifications might be short lived, and further dose-finding studies will be required to examine the optimal dosing and duration of the formula. Second, the duration of follow up lasted for three months only, and continuous observation of these subjects is warranted to assess the sustainability of these beneficial effects. Furthermore, our study requires it to be repeated in other regions as microbiome composition may vary in subjects of different ethnicities. Lastly, the questionnaire items to assess the quality of life were evaluated for face validity only, and the findings should be interpreted with caution.

In conclusion, our findings demonstrated a significant role of the novel microbiome formula, SIM01, in reducing adverse health outcomes in elderly subjects and patients with type two diabetes mellitus, which was associated with the restoration of gut dysbiosis. These findings provide significant societal implications for strategies that could protect these vulnerable individuals during the COVID-19 pandemic. We recommend that future research should compare different measurable markers between the SIM01 group and the control group.

## Figures and Tables

**Figure 1 nutrients-15-01982-f001:**
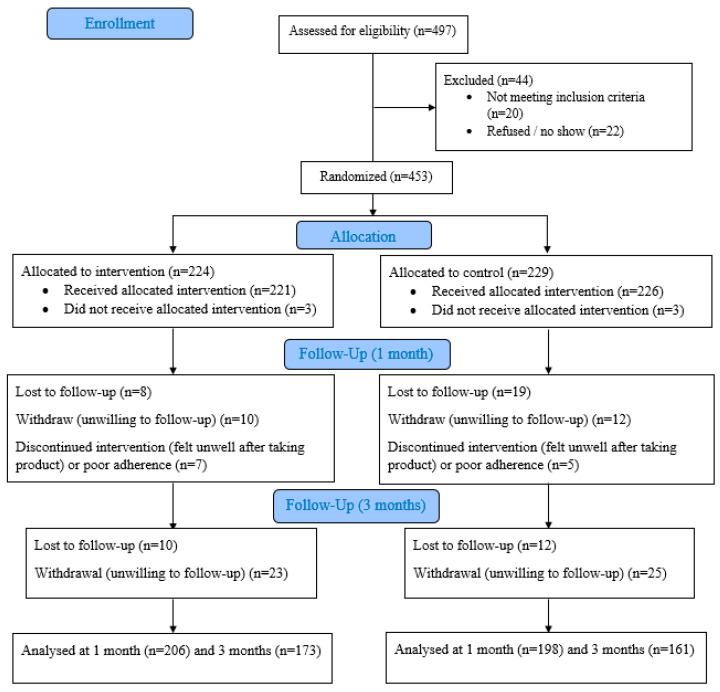
CONSORT Flow diagram.

**Figure 2 nutrients-15-01982-f002:**
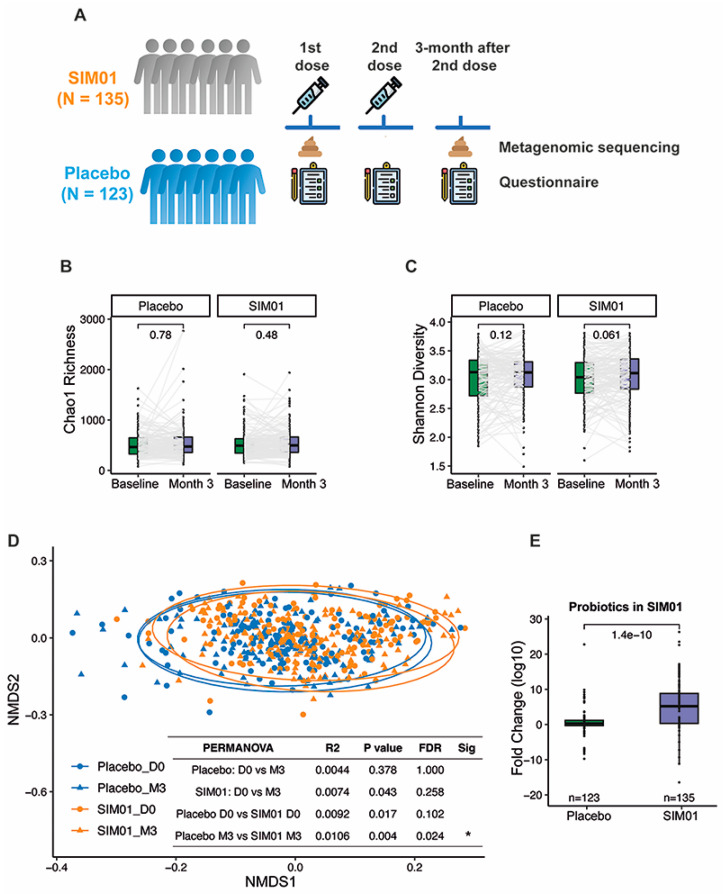
Study design and changes in alpha diversity and beta diversity from baseline to 3 months after the second dose of vaccination. (**A**) Study design. (**B**) Chao1 richness between baseline and 3 months after the second vaccine dose within each group. (**C**) SIM01 oral administration marginally increased the species level-Shannon diversity at 3 months after the second vaccine dose versus the baseline (placebo baseline, *n* = 123; placebo 3 months, *n* = 123; SIM01 baseline, *n* = 135; SIM01 3 months, *n* = 135). *p* values were given by paired Wilcoxon rank-sum test (two-sided). Elements on boxplots: centre line, median; box limits, upper and lower quartiles; points, each sample; grey line, the same paired subject. (**D**) Beta diversity was significantly different between placebo and SIM01 groups at month 3 after the 2nd dose of vaccination (placebo baseline, *n* = 123; placebo 3 months, *n* = 123; SIM01 baseline, *n* = 135; SIM01 3 months, *n* = 135). Pairwise multilevel comparisons were carried out on the JSD distance matrix using a pairwise Adonis test. (**E**) The fold change (month 3/baseline) of three probiotic species in SIM01 was higher in the SIM01 group compared to the placebo group. The *p* values were given by the Wilcoxon rank-sum test (two-sided); *p* values were given by PERMANOVA and adjusted for FDR, respectively. FDR, false discovery rate; NMDS, nonmetric multidimensional scaling; PERMANOVA, permutational multivariate analysis of variance; JSD, Jensen-Shannon Divergence (distance); * indicates FDR < 0.05.

**Figure 3 nutrients-15-01982-f003:**
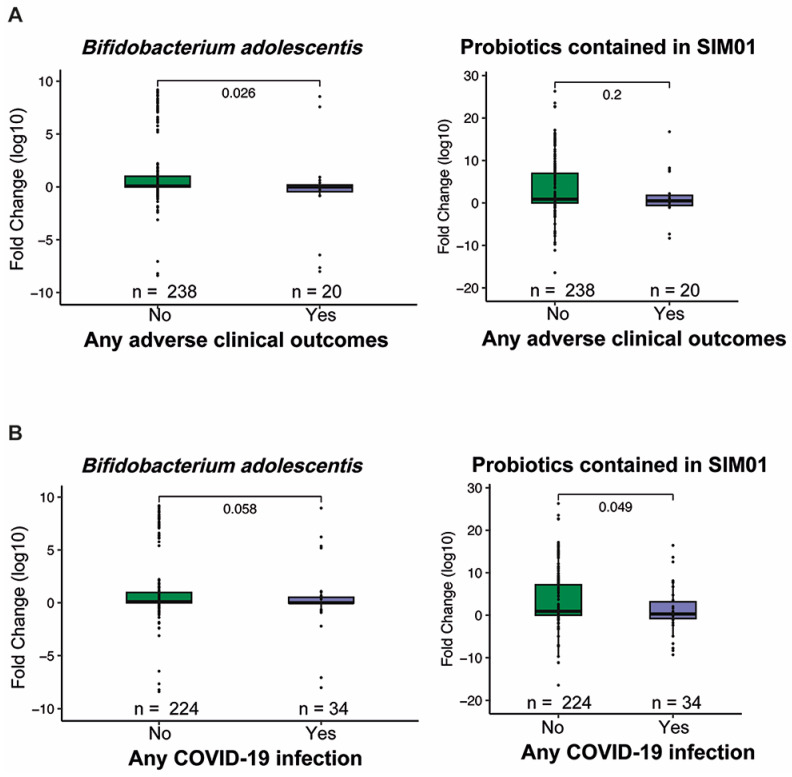
(**A**) The fold change (month 3/baseline) of *B. adolescentis* was lower in subjects with any adverse health outcomes. (**B**) The fold change (month 3/baseline) of *B. adolescentis* and the fold change (month 3/baseline) of three probiotic species in SIM01 were lower in subjects with COVID-19 infection. The *p* values were given by the Wilcoxon rank-sum test (two-sided). Elements on boxplots: centre line, median; box limits, upper and lower quartiles; points, each sample.

**Figure 4 nutrients-15-01982-f004:**
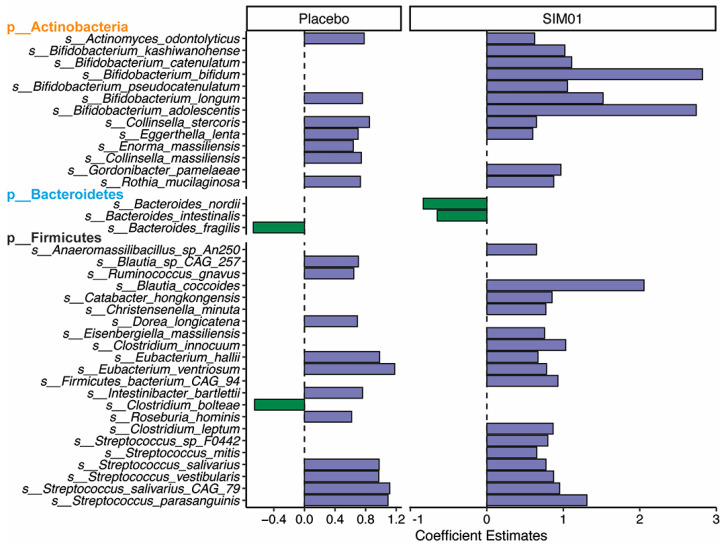
Differential species between baseline and 3 months after the second dose of vaccination for placebo (*n* = 123) and SIM01 (*n* = 135). Differentially abundant species were detected using Maasline2 considering the subjects as random effects (*p* value < 0.05, FDR corrected *p* < 0.25). Purple colour, species enriched during 3 months after the second dose of vaccination versus the baseline; green colour, species depleted during 3 months after the second dose of vaccination versus the baseline.

**Figure 5 nutrients-15-01982-f005:**
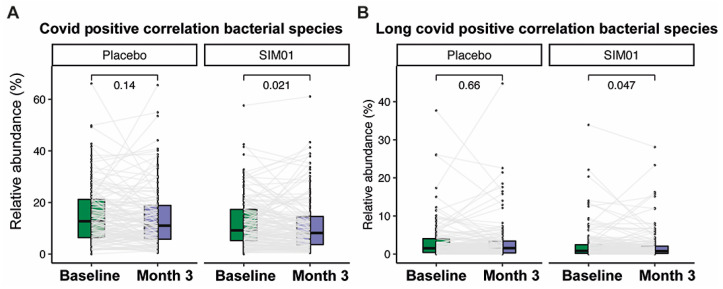
The abundance of pathogenic bacterial species enriched in COVID-19 or Long COVID was significantly decreased at month 3 in the SIM01 arm (placebo baseline, *n* = 123; placebo 3 months, *n* = 123; SIM01 baseline, *n* = 135; SIM01 3 month, *n* = 135). Elements on boxplots: centre line, median; box limits, upper and lower quartiles; points, each sample; grey line, the same paired subject.

**Figure 6 nutrients-15-01982-f006:**
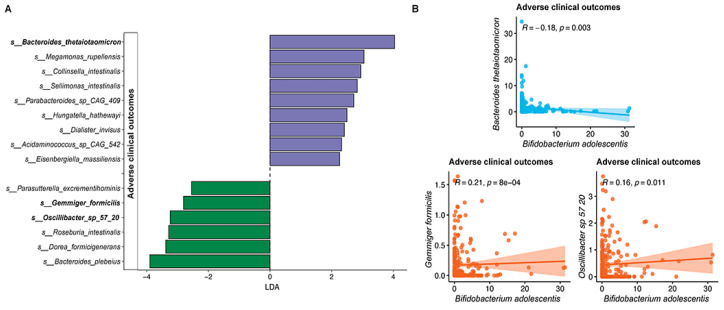
(**A**) Differential species between subjects with (*n* = 20) or without any adverse health outcomes (*n* = 238) at 3 months after the second dose of vaccination. Differentially abundant species groups were identified using the linear discriminant analysis effect size (LEfSe) (linear discriminant analysis (LDA) score > 2, and *p* < 0.05). Purple colour, species enriched in subjects with adverse health outcomes; green colour, species depleted in subjects with adverse health outcomes. (**B**) Correlation between *Bifidobacterium adolescentis* with species associated with adverse health outcomes. The correlation between the species abundance was analyzed using Spearman’s correlation tests.

**Table 1 nutrients-15-01982-t001:** Participant characteristics at baseline.

	Microbiome Immunity Formula SIM01 (*n* = 224)	Control (*n* = 229)	All Subjects (*n* = 453)
Age (mean, SD), years	67.4 (8.3)	67.6 (8.0)	67.5 (8.1)
Sex (*n*, %)			
Female	107 (47.8)	121 (52.8)	228 (50.3)
Male	117 (52.2)	108 (47.2)	225 (49.7)
Body weight (mean; SD), kg	64.0 (13.6)	62.9 (13.2)	63.7 (13.5)
Smoking status (*n*, %)			
Current	13 (5.8)	15 (6.6)	28 (6.2)
Ex-smoker	26 (11.6)	15 (6.6)	41 (9.1)
Nonsmoker	185 (82.6)	199 (86.8)	384 (84.7)
Alcohol drinking (*n*, %)			
Current drinker	31 (13.84)	17 (7.42)	48 (10.60)
Exdrinker	7 (3.13)	11 (4.81)	18 (3.97)
Nondrinker	186 (83.0)	201 (87.8)	387 (85.4)
Chronic disease (*n*, %)			
Hypertension	115 (51.3)	108 (47.1)	223 (49.2)
Diabetes Mellitus	111 (49.5)	104 (45.4)	215 (47.5)
Hyperlipidaemia	108 (48.2)	105 (45.8)	213 (47)
Fatty Liver	9 (4)	15 (6.6)	24 (5.3)
Gout	8 (3.5)	9 (3.9)	17 (3.8)
Cardiovascular Disease	17 (7.6)	21 (9.2)	38 (8.4)
OSAS	6 (2.7)	10 (4.4)	16 (3.5)
Asthma	9 (4)	5 (2.2)	14 (3.1)
CVA	2 (0.9)	4 (1.7)	6 (1.3)
Use of chronic medications (*n*, %)			
Antihypertensive agents	103 (46.0)	104 (45.4)	207 (45.7)
Oral hypoglycemic agents/insulin	76 (33.9)	66 (28.8)	142 (31.3)
Lipid-lowering agents	97 (43.3)	94 (41.0)	191 (42.2)
Antiplatelets/anticoagulants	39 (17.4)	39 (17.0)	78 (17.2)
H_2_ receptor antagomist/PPI	58 (25.9)	59 (25.8)	117 (25.8)
Others *	97 (43.3)	93 (40.6)	190 (41.9)

OSAS: Obstructive Sleep Apnea Syndrome; CVA: Cerebrovascular Accident. PPI: Proton Pump Inhibitors. * include chronic medications for diseases of other bodily systems.

**Table 2 nutrients-15-01982-t002:** The incidence of adverse health outcomes at 1 month and 3 months.

	Month 1	Month 3
Microbiome immunity formula SIM01	6/206 (2.9%)	0/173 (0%)
	Gastrointestinal: Bloating (1) Constipation (3) Diarrhoea (1) Immunological: Dermatitis (1)	
Vitamin C	25/198 (12.6%) Gastrointestinal: Bloating (3) Constipation (5) Diarrhoea (3) Abdominal discomfort (2) Loss of appetite (1) Immunological: Eczematous rash (2) Urticarial rash (1) Allergic skin reaction (2) Infection: Infected liver cyst (1) Infected wound (4) Septic shock (1)	5/161 (3.1%) Gastrointestinal: Constipation (2) Infection: Wound infection (2) COVID-19 (1)
*p* value		
Overall	0.0002 *	0.0252 *

* *p* < 0.05.

## Data Availability

Anonymised data reported in the manuscript will be made available to investigators who provide a methodologically sound proposal to the corresponding author. The protocol is available upon request.

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
