# Peer review of "Effects of Gut Microbiome Modulation on Reducing Adverse Health Outcomes among Elderly and Diabetes Patients during the COVID-19 Pandemic: A Randomised, Double-Blind, Placebo-Controlled Trial (IMPACT Study)"

_nutrients, 2023, doi:10.3390/nu15081982_

Round 1

Reviewer 1 Report (Previous Reviewer 2)

The MS has been revised as per the previous comments. Authors used SIM01 is an oral microencapsulated formulation that contain probiotic Bifidobacteria. Context to this it would be better if author can add few lines concerning prebiotics such as xylooligosaccharide (XOS) in the introduction section. Author can refer the reference : https://doi.org/10.1016/j.jclepro.2021.128332 for the role prebiotics.  

Author Response

Ms. Julie Zhu

Assistant Editor,

The journal “Nutrients”

17 April, 2023

Re: Resubmission of the article entitled “Effects of gut microbiome modulation on reducing adverse health outcomes among elderly and diabetes patients during the COVID-19 pandemic: a randomised, double-blind, placebo-controlled trial (IMPACT Study)”

Dear Julie,

Thank you very much for your message offering a precious opportunity for revision of the captioned manuscript. We have now revised the article and listed our responses point-by-point as contained in this letter.

Your further advice on this revised article will be highly appreciated. Thank you very much and we look forward to your kind comments.

Sincerely,

Professor Martin CS Wong

The JC School of Public Health and Primary Care, The Chinese University of Hong Kong

Tel: (852) 2252-8782; Fax: (852) 2606-3500; Email: [email protected];

Address: Room 407, 4/F, JC School of Public Health and Primary Care, Prince of Wales Hospital, Shatin, NT, Hong Kong SAR

Editorial comments

English editing requested for the manuscript.

Our response: Thank you. The manuscript was further edited by an expert in English language.

Reviewer 1

The MS has been revised as per the previous comments. Authors used SIM01 is an oral microencapsulated formulation that contain probiotic Bifidobacteria. Context to this it would be better if author can add few lines concerning prebiotics such as xylooligosaccharide (XOS) in the introduction section. Author can refer the reference: https://doi.org/10.1016/j.jclepro.2021.128332 for the role prebiotics. 

Our response: Thank you for your comments. The Introduction section has now included prebiotics such as xylooligosaccharide (XOS) to contextualize the background of the study. We have also included the reference provided in the article. We have added the following paragraph:

 “In the past decade, major prebiotics compounds such as xylooligosaccharides were found to be effective in enhancing the growth and activity of gut symbionts [15], such as Lactobacillus, Bifidobacterium, and Anaerostipes spp. In addition, studies have shown that probiotics regulate the functions of systemic and mucosal immune cells and intestinal epithelial cells [16].”

Please see lines 62-66 for the revision.

Reviewer 2

The manuscript reports the effects of a bacterial formulation on elderly and diabetic subjects during the COVID period. What is the exact content of single bacterial species in the formulation?

Our response: We have further included details of the formulation content. The added paragraph reads: “SIM01 is an oral microencapsulated formulation of three lyophilised Bifidobacteria at a dose of 20 billion Colony Forming Units (CFU) per day and 3 prebiotics prebiotics, including galactooligosaccharides, xylooligosaccharide, and resistant dextrin as derived from our previous study. The three Bifidobacterium strains (B adolescentis; B bididum, and B longum) are commercially available from Chambio Co., Ltd and WECARE-PROBIOTICS. SIM01 has been approved by health authorities and is a patent-protected formula.” Please see lines 113-118 for the revision.

line 100, all subjects are over 65, and it is therefore superfluous to report ”...were18 years or above.”

Our response: We agree, the sentence was deleted. Please see lines 103-104 for the revision.

Subjects take various drugs. This should be entered.  

Our response: Thank you for this comment. We have now included the various types of medications taken by our study participants in Table 1.

Uniform definition of formulation and placebo within the text.

Our response: We have presented uniform definitions of the formulation (the “SIM01 group”) and placebo (“placebo group”) throughout the text. Thank you.

The discussion should be expanded in light of the extensive literature on microbiota, probiotics and covid.

Our response: Thank you for this comment. We have now expanded the Discussion section by including the following paragraph which linked microbiota, probiotics and COVID-19 together in the context of the present study. It reads:

“Observational studies have shown that patients with COVID-19 infection reported altered gut microbiota composition, including reduced bacterial diversity, decreased abundance of short-chain fatty acid-producing bacteria from the Lachnospiraceae, Ruminococcaceae and Eubacteriaceae families, as well as increased opportunistic pathogens from the Enterobacteriaceae families. Nevertheless, oral administration of oral probiotics and prebiotics has been proven effective to induce antiviral activity and exert favourable effects on the gut microbial diversity and composition which was associated with better clinical outcomes in COVID-19 patients. Furthermore, a recent study in the US by Wischmeyer and col-leagues examined the probiotic Lactobacillus rhamnosus GG (LGG) as postexposure prophylaxis for COVID-19 in 182 participants who had household exposure to some-one with confirmed COVID-19 diagnosed within ≤ 7 days. The findings suggested that LGG was associated with prolonged time to COVID-19 infection, reduced incidence of illness symptoms, and gut microbiome changes. However, the study was limited by a smaller-than-expected sample size, which limited its statistical power. Secondly, this US study has not included vaccinated individuals, which is an important limitation given the high transmissibility of newer viral strains and potential for waning vaccine efficacy.”

Please see lines 368-384 for the revision.

Ms. Julie Zhu

Assistant Editor,

The journal “Nutrients”

17 April, 2023

Re: Resubmission of the article entitled “Effects of gut microbiome modulation on reducing adverse health outcomes among elderly and diabetes patients during the COVID-19 pandemic: a randomised, double-blind, placebo-controlled trial (IMPACT Study)”

Dear Julie,

Thank you very much for your message offering a precious opportunity for revision of the captioned manuscript. We have now revised the article and listed our responses point-by-point as contained in this letter.

Your further advice on this revised article will be highly appreciated. Thank you very much and we look forward to your kind comments.

Sincerely,

Professor Martin CS Wong

The JC School of Public Health and Primary Care, The Chinese University of Hong Kong

Tel: (852) 2252-8782; Fax: (852) 2606-3500; Email: [email protected];

Address: Room 407, 4/F, JC School of Public Health and Primary Care, Prince of Wales Hospital, Shatin, NT, Hong Kong SAR

Editorial comments

English editing requested for the manuscript.

Our response: Thank you. The manuscript was further edited by an expert in English language.

Reviewer 1

The MS has been revised as per the previous comments. Authors used SIM01 is an oral microencapsulated formulation that contain probiotic Bifidobacteria. Context to this it would be better if author can add few lines concerning prebiotics such as xylooligosaccharide (XOS) in the introduction section. Author can refer the reference: https://doi.org/10.1016/j.jclepro.2021.128332 for the role prebiotics. 

Our response: Thank you for your comments. The Introduction section has now included prebiotics such as xylooligosaccharide (XOS) to contextualize the background of the study. We have also included the reference provided in the article. We have added the following paragraph:

 “In the past decade, major prebiotics compounds such as xylooligosaccharides were found to be effective in enhancing the growth and activity of gut symbionts [15], such as Lactobacillus, Bifidobacterium, and Anaerostipes spp. In addition, studies have shown that probiotics regulate the functions of systemic and mucosal immune cells and intestinal epithelial cells [16].”

Please see lines 62-66 for the revision.

Reviewer 2

The manuscript reports the effects of a bacterial formulation on elderly and diabetic subjects during the COVID period. What is the exact content of single bacterial species in the formulation?

Our response: We have further included details of the formulation content. The added paragraph reads: “SIM01 is an oral microencapsulated formulation of three lyophilised Bifidobacteria at a dose of 20 billion Colony Forming Units (CFU) per day and 3 prebiotics prebiotics, including galactooligosaccharides, xylooligosaccharide, and resistant dextrin as derived from our previous study. The three Bifidobacterium strains (B adolescentis; B bididum, and B longum) are commercially available from Chambio Co., Ltd and WECARE-PROBIOTICS. SIM01 has been approved by health authorities and is a patent-protected formula.” Please see lines 113-118 for the revision.

line 100, all subjects are over 65, and it is therefore superfluous to report ”...were18 years or above.”

Our response: We agree, the sentence was deleted. Please see lines 103-104 for the revision.

Subjects take various drugs. This should be entered.  

Our response: Thank you for this comment. We have now included the various types of medications taken by our study participants in Table 1.

Uniform definition of formulation and placebo within the text.

Our response: We have presented uniform definitions of the formulation (the “SIM01 group”) and placebo (“placebo group”) throughout the text. Thank you.

The discussion should be expanded in light of the extensive literature on microbiota, probiotics and covid.

Our response: Thank you for this comment. We have now expanded the Discussion section by including the following paragraph which linked microbiota, probiotics and COVID-19 together in the context of the present study. It reads:

“Observational studies have shown that patients with COVID-19 infection reported altered gut microbiota composition, including reduced bacterial diversity, decreased abundance of short-chain fatty acid-producing bacteria from the Lachnospiraceae, Ruminococcaceae and Eubacteriaceae families, as well as increased opportunistic pathogens from the Enterobacteriaceae families. Nevertheless, oral administration of oral probiotics and prebiotics has been proven effective to induce antiviral activity and exert favourable effects on the gut microbial diversity and composition which was associated with better clinical outcomes in COVID-19 patients. Furthermore, a recent study in the US by Wischmeyer and col-leagues examined the probiotic Lactobacillus rhamnosus GG (LGG) as postexposure prophylaxis for COVID-19 in 182 participants who had household exposure to some-one with confirmed COVID-19 diagnosed within ≤ 7 days. The findings suggested that LGG was associated with prolonged time to COVID-19 infection, reduced incidence of illness symptoms, and gut microbiome changes. However, the study was limited by a smaller-than-expected sample size, which limited its statistical power. Secondly, this US study has not included vaccinated individuals, which is an important limitation given the high transmissibility of newer viral strains and potential for waning vaccine efficacy.”

Please see lines 368-384 for the revision.

Ms. Julie Zhu

Assistant Editor,

The journal “Nutrients”

17 April, 2023

Re: Resubmission of the article entitled “Effects of gut microbiome modulation on reducing adverse health outcomes among elderly and diabetes patients during the COVID-19 pandemic: a randomised, double-blind, placebo-controlled trial (IMPACT Study)”

Dear Julie,

Thank you very much for your message offering a precious opportunity for revision of the captioned manuscript. We have now revised the article and listed our responses point-by-point as contained in this letter.

Your further advice on this revised article will be highly appreciated. Thank you very much and we look forward to your kind comments.

Sincerely,

Professor Martin CS Wong

The JC School of Public Health and Primary Care, The Chinese University of Hong Kong

Tel: (852) 2252-8782; Fax: (852) 2606-3500; Email: [email protected];

Address: Room 407, 4/F, JC School of Public Health and Primary Care, Prince of Wales Hospital, Shatin, NT, Hong Kong SAR

Editorial comments

English editing requested for the manuscript.

Our response: Thank you. The manuscript was further edited by an expert in English language.

Reviewer 1

The MS has been revised as per the previous comments. Authors used SIM01 is an oral microencapsulated formulation that contain probiotic Bifidobacteria. Context to this it would be better if author can add few lines concerning prebiotics such as xylooligosaccharide (XOS) in the introduction section. Author can refer the reference: https://doi.org/10.1016/j.jclepro.2021.128332 for the role prebiotics. 

Our response: Thank you for your comments. The Introduction section has now included prebiotics such as xylooligosaccharide (XOS) to contextualize the background of the study. We have also included the reference provided in the article. We have added the following paragraph:

 “In the past decade, major prebiotics compounds such as xylooligosaccharides were found to be effective in enhancing the growth and activity of gut symbionts [15], such as Lactobacillus, Bifidobacterium, and Anaerostipes spp. In addition, studies have shown that probiotics regulate the functions of systemic and mucosal immune cells and intestinal epithelial cells [16].”

Please see lines 62-66 for the revision.

Reviewer 2

The manuscript reports the effects of a bacterial formulation on elderly and diabetic subjects during the COVID period. What is the exact content of single bacterial species in the formulation?

Our response: We have further included details of the formulation content. The added paragraph reads: “SIM01 is an oral microencapsulated formulation of three lyophilised Bifidobacteria at a dose of 20 billion Colony Forming Units (CFU) per day and 3 prebiotics prebiotics, including galactooligosaccharides, xylooligosaccharide, and resistant dextrin as derived from our previous study. The three Bifidobacterium strains (B adolescentis; B bididum, and B longum) are commercially available from Chambio Co., Ltd and WECARE-PROBIOTICS. SIM01 has been approved by health authorities and is a patent-protected formula.” Please see lines 113-118 for the revision.

line 100, all subjects are over 65, and it is therefore superfluous to report ”...were18 years or above.”

Our response: We agree, the sentence was deleted. Please see lines 103-104 for the revision.

Subjects take various drugs. This should be entered.  

Our response: Thank you for this comment. We have now included the various types of medications taken by our study participants in Table 1.

Uniform definition of formulation and placebo within the text.

Our response: We have presented uniform definitions of the formulation (the “SIM01 group”) and placebo (“placebo group”) throughout the text. Thank you.

The discussion should be expanded in light of the extensive literature on microbiota, probiotics and covid.

Our response: Thank you for this comment. We have now expanded the Discussion section by including the following paragraph which linked microbiota, probiotics and COVID-19 together in the context of the present study. It reads:

“Observational studies have shown that patients with COVID-19 infection reported altered gut microbiota composition, including reduced bacterial diversity, decreased abundance of short-chain fatty acid-producing bacteria from the Lachnospiraceae, Ruminococcaceae and Eubacteriaceae families, as well as increased opportunistic pathogens from the Enterobacteriaceae families. Nevertheless, oral administration of oral probiotics and prebiotics has been proven effective to induce antiviral activity and exert favourable effects on the gut microbial diversity and composition which was associated with better clinical outcomes in COVID-19 patients. Furthermore, a recent study in the US by Wischmeyer and col-leagues examined the probiotic Lactobacillus rhamnosus GG (LGG) as postexposure prophylaxis for COVID-19 in 182 participants who had household exposure to some-one with confirmed COVID-19 diagnosed within ≤ 7 days. The findings suggested that LGG was associated with prolonged time to COVID-19 infection, reduced incidence of illness symptoms, and gut microbiome changes. However, the study was limited by a smaller-than-expected sample size, which limited its statistical power. Secondly, this US study has not included vaccinated individuals, which is an important limitation given the high transmissibility of newer viral strains and potential for waning vaccine efficacy.”

Please see lines 368-384 for the revision.

Reviewer 2 Report (New Reviewer)

The manuscript reports the effects of a bacterial formulation on elderly and diabetic subjects during the COVID period.

What is the exact content of single bacterial species in the formulation? line 100, all subjects are over 65, and it is therefore superfluous to report ”...were18 years or above.” Subjects take various drugs. This should be entered.   Uniform definition of formulation and placebo within the text.
The discussion should be expanded in light of the extensive literature on microbiota, probiotics and covid.

Author Response

Ms. Julie Zhu

Assistant Editor,

The journal “Nutrients”

17 April, 2023

Re: Resubmission of the article entitled “Effects of gut microbiome modulation on reducing adverse health outcomes among elderly and diabetes patients during the COVID-19 pandemic: a randomised, double-blind, placebo-controlled trial (IMPACT Study)”

Dear Julie,

Thank you very much for your message offering a precious opportunity for revision of the captioned manuscript. We have now revised the article and listed our responses point-by-point as contained in this letter.

Your further advice on this revised article will be highly appreciated. Thank you very much and we look forward to your kind comments.

Sincerely,

Professor Martin CS Wong

The JC School of Public Health and Primary Care, The Chinese University of Hong Kong

Tel: (852) 2252-8782; Fax: (852) 2606-3500; Email: [email protected];

Address: Room 407, 4/F, JC School of Public Health and Primary Care, Prince of Wales Hospital, Shatin, NT, Hong Kong SAR

Editorial comments

English editing requested for the manuscript.

Our response: Thank you. The manuscript was further edited by an expert in English language.

Reviewer 1

The MS has been revised as per the previous comments. Authors used SIM01 is an oral microencapsulated formulation that contain probiotic Bifidobacteria. Context to this it would be better if author can add few lines concerning prebiotics such as xylooligosaccharide (XOS) in the introduction section. Author can refer the reference: https://doi.org/10.1016/j.jclepro.2021.128332 for the role prebiotics. 

Our response: Thank you for your comments. The Introduction section has now included prebiotics such as xylooligosaccharide (XOS) to contextualize the background of the study. We have also included the reference provided in the article. We have added the following paragraph:

 “In the past decade, major prebiotics compounds such as xylooligosaccharides were found to be effective in enhancing the growth and activity of gut symbionts [15], such as Lactobacillus, Bifidobacterium, and Anaerostipes spp. In addition, studies have shown that probiotics regulate the functions of systemic and mucosal immune cells and intestinal epithelial cells [16].”

Please see lines 62-66 for the revision.

Reviewer 2

The manuscript reports the effects of a bacterial formulation on elderly and diabetic subjects during the COVID period. What is the exact content of single bacterial species in the formulation?

Our response: We have further included details of the formulation content. The added paragraph reads: “SIM01 is an oral microencapsulated formulation of three lyophilised Bifidobacteria at a dose of 20 billion Colony Forming Units (CFU) per day and 3 prebiotics prebiotics, including galactooligosaccharides, xylooligosaccharide, and resistant dextrin as derived from our previous study. The three Bifidobacterium strains (B adolescentis; B bididum, and B longum) are commercially available from Chambio Co., Ltd and WECARE-PROBIOTICS. SIM01 has been approved by health authorities and is a patent-protected formula.” Please see lines 113-118 for the revision.

line 100, all subjects are over 65, and it is therefore superfluous to report ”...were18 years or above.”

Our response: We agree, the sentence was deleted. Please see lines 103-104 for the revision.

Subjects take various drugs. This should be entered.  

Our response: Thank you for this comment. We have now included the various types of medications taken by our study participants in Table 1.

Uniform definition of formulation and placebo within the text.

Our response: We have presented uniform definitions of the formulation (the “SIM01 group”) and placebo (“placebo group”) throughout the text. Thank you.

The discussion should be expanded in light of the extensive literature on microbiota, probiotics and covid.

Our response: Thank you for this comment. We have now expanded the Discussion section by including the following paragraph which linked microbiota, probiotics and COVID-19 together in the context of the present study. It reads:

“Observational studies have shown that patients with COVID-19 infection reported altered gut microbiota composition, including reduced bacterial diversity, decreased abundance of short-chain fatty acid-producing bacteria from the Lachnospiraceae, Ruminococcaceae and Eubacteriaceae families, as well as increased opportunistic pathogens from the Enterobacteriaceae families. Nevertheless, oral administration of oral probiotics and prebiotics has been proven effective to induce antiviral activity and exert favourable effects on the gut microbial diversity and composition which was associated with better clinical outcomes in COVID-19 patients. Furthermore, a recent study in the US by Wischmeyer and col-leagues examined the probiotic Lactobacillus rhamnosus GG (LGG) as postexposure prophylaxis for COVID-19 in 182 participants who had household exposure to some-one with confirmed COVID-19 diagnosed within ≤ 7 days. The findings suggested that LGG was associated with prolonged time to COVID-19 infection, reduced incidence of illness symptoms, and gut microbiome changes. However, the study was limited by a smaller-than-expected sample size, which limited its statistical power. Secondly, this US study has not included vaccinated individuals, which is an important limitation given the high transmissibility of newer viral strains and potential for waning vaccine efficacy.”

Please see lines 368-384 for the revision.

This manuscript is a resubmission of an earlier submission. The following is a list of the peer review reports and author responses from that submission.

Round 1

Reviewer 1 Report

The introduction is well done. The material and method are well explained and the selection and allocation of patients is well done.

The authors showed a modification of the microbiota after consumption of SIMO1. These results are interesting because a real increase of bifidobacteria is observed, of which those belonging to the SIM01 composition have been demonstrated. The authors explained an improvement of the adverse effects.

However, the beneficial effects on the improvement of adverse events of Covid19 vaccinations are explained by the authors but not really demonstrated. Measurable markers of these improvements are missing.

Author Response

The introduction is well done. The material and method are well explained and the selection and allocation of patients is well done. The authors showed a modification of the microbiota after consumption of SIMO1. These results are interesting because a real increase of bifidobacteria is observed, of which those belonging to the SIM01 composition have been demonstrated. The authors explained an improvement of the adverse effects.

Our response: We are grateful for the positive comments from the reviewer.

However, the beneficial effects on the improvement of adverse events of Covid19 vaccinations are explained by the authors but not really demonstrated. Measurable markers of these improvements are missing.

Our response: We have performed a subgroup analysis and compared the incidence of possible adverse events of COVID-19 vaccination. The incidence was 2.9% (6/206) in the SIM01 group and 9.6% (19/198) in the control group at 1 month. The corresponding incidence rate was 0% (0/173) in the SIM01 group and 1.2% (2/161) at 3 months. Please see lines 254-255 for the inclusion of this additional findings.

We agree with the reviewer that other markers could also be measured to compare the difference between the SIM01 group and the control group. We have included this as a recommendation for future research in the Discussion section. Please see lines 373-375 for our revision.

Reviewer 2 Report

The manuscript “Effects of gut microbiome modulation on health outcomes among elderly and diabetes patients: a randomized, double-blind, placebo-controlled trial (IMPACT Study)”. Any natural things with nutritional and medicinal properties are highly appreciable. The study is good and the findings are satisfactory. The study has a practical utility as novel oral microbiome formula (SIM01) on gut health as demonstrating the positive effect on human health during trials.

  1. In the text, section add information about the total number of participants in the study. 
  2. Reference 17 (line 157) should be present as per the guidance of the journal.
  3. All the references cited in the text should be placed before the full stop. Please check the guidelines of the journal. 
  4. The full name of the abbreviation such as DM should be added while appearing first in the text or table.
  5. Delete the word “Study”, “Potentially” and “Our” from line 363 of the conclusion part. 

Author Response

The manuscript “Effects of gut microbiome modulation on health outcomes among elderly and diabetes patients: a randomized, double-blind, placebo-controlled trial (IMPACT Study)”. Any natural things with nutritional and medicinal properties are highly appreciable. The study is good and the findings are satisfactory. The study has a practical utility as novel oral microbiome formula (SIM01) on gut health as demonstrating the positive effect on human health during trials.

Our response: We are grateful for the positive comments from the reviewer.

In the text, section add information about the total number of participants in the study.

Our response: We have included the total number of study participants in this study. We have screened 497 subjects and enrolled 453 eligible subjects in the trial. Please see lines 228 and 231.

Reference 17 (line 157) should be present as per the guidance of the journal. All the references cited in the text should be placed before the full stop. Please check the guidelines of the journal.

Our response: We have positioned all the references in the text as suggested by the reviewer (i.e. the references cited are place before the full stop, including reference 17). Thank you.

The full name of the abbreviation such as DM should be added while appearing first in the text or table.

Our response: We have checked for all the abbreviations and ensured their full spellings appear first in the text and Tables.

Delete the word “Study”, “Potentially” and “Our” from line 363 of the conclusion part.

Our response: We have deleted all these words as suggested by the reviewer (now line 369-370)